# *Escherichia coli* Is Overtaking Group B *Streptococcus* in Early-Onset Neonatal Sepsis

**DOI:** 10.3390/microorganisms10101878

**Published:** 2022-09-20

**Authors:** Francesca Miselli, Riccardo Cuoghi Costantini, Roberta Creti, Francesca Sforza, Silvia Fanaro, Matilde Ciccia, Giancarlo Piccinini, Vittoria Rizzo, Lorena Pasini, Giacomo Biasucci, Rossella Pagano, Mariagrazia Capretti, Mariachiara China, Lucia Gambini, Rita Maria Pulvirenti, Arianna Dondi, Marcello Lanari, MariaFederica Pedna, Simone Ambretti, Licia Lugli, Luca Bedetti, Alberto Berardi

**Affiliations:** 1Neonatal Intensive Care Unit, Women’s and Children’s Health Department, University Hospital of Modena, 41124 Modena, Italy; 2Department of Medical and Surgical Sciences for Mother, Child and Adult, University of Modena and Reggio Emilia, 41124 Modena, Italy; 3Department of Infectious Diseases, Istituto Superiore di Sanità, 00161 Rome, Italy; 4Pediatric Post-Graduate School, University of Modena e Reggio Emilia, 41125 Modena, Italy; 5Department of Medical Sciences, Pediatric Section, University of Ferrara, 44124 Ferrara, Italy; 6Neonatal Intensive Care Unit, Women’s and Children’s Health Department, Maggiore Hospital, 40133 Bologna, Italy; 7Pediatric Unit, Santa Maria Delle Croci Hospital, 48121 Ravenna, Italy; 8Neonatal Intensive Care Unit, Bufalini Hospital of Cesena, 47521 Cesena, Italy; 9Department of Paediatric Anaesthesia and Intensive Care, S.Orsola-Malpighi Hospital, 40138 Bologna, Italy; 10Pediatric and Neonatal Unit, Women’s and Children’s Health Department, Guglielmo da Saliceto Hospital, 29121 Piacenza, Italy; 11Pediatric Unit, Hospital of Sassuolo, 41049 Sassuolo, Italy; 12Neonatal Intensive Care Unit, Women’s and Children’s Health Department, S.Orsola-Malpighi Hospital, 40138 Bologna, Italy; 13Neonatal Intensive Care Unit, Infermi Hospital, 47923 Rimini, Italy; 14Neonatal Intensive Care Unit, University Hospital of Parma, 43126 Parma, Italy; 15Pediatric and Neonatal Unit, Morgagni-Pierantoni Hospital of Forlì, 47121 Forlì, Italy; 16Pediatric Emergency Unit, Scientific Institute for Research and Healthcare (IRCCS), Sant’Orsola Hospital, 40138 Bologna, Italy; 17Unit of Microbiology, Romagna Laboratory Unit, 47522 Cesena, Italy; 18Unit of Clinical Microbiology, Scientific Institute for Research and Healthcare (IRCCS), Sant’Orsola Hospital, 40138 Bologna, Italy; 19PhD Program in Clinical and Experimental Medicine, University of Modena and Reggio Emilia, 41121 Modena, Italy

**Keywords:** early-onset sepsis, neonatal sepsis, group B *streptococcus*, *Escherichia coli*, intrapartum antibiotic prophylaxis, epidemiology, antibiotic resistance

## Abstract

The widespread use of intrapartum antibiotic prophylaxis (IAP) to prevent group B *streptococcus* (GBS) early-onset sepsis (EOS) is changing the epidemiology of EOS. Italian prospective area-based surveillance data (from 1 January 2016 to 31 December 2020) were used, from which we identified 64 cases of culture-proven EOS (*E. coli*, *n* = 39; GBS, *n* = 25) among 159,898 live births (annual incidence rates of 0.24 and 0.16 per 1000, respectively). Approximately 10% of *E. coli* isolates were resistant to both gentamicin and ampicillin. Five neonates died; among them, four were born very pre-term (*E. coli*, *n* = 3; GBS, *n* = 1) and one was born full-term (*E. coli*, n = 1). After adjustment for gestational age, IAP-exposed neonates had ≥95% lower risk of death, as compared to IAP-unexposed neonates, both in the whole cohort (OR 0.04, 95% CI 0.00–0.70; *p* = 0.03) and in the *E. coli* EOS cohort (OR 0.05, 95% CI 0.00–0.88; *p* = 0.04). In multi-variable logistic regression analysis, IAP was inversely associated with severe disease (OR = 0.12, 95% CI 0.02–0.76; *p* = 0.03). *E. coli* is now the leading pathogen in neonatal EOS, and its incidence is close to that of GBS in full-term neonates. IAP reduces the risk of severe disease and death. Importantly, approximately 10% of *E. coli* isolates causing EOS were found to be resistant to typical first-line antibiotics.

## 1. Introduction

Early-onset sepsis (EOS) is still a serious and potentially fatal neonatal complication, particularly among neonates of the lowest gestational age; as such, identifying and managing infants at risk for EOS is part of the routine care provided by neonatal clinicians. 

The epidemiology of neonatal sepsis is a changing landscape [1,2]. In particular, the widespread implementation of intrapartum antibiotic prophylaxis (IAP) to reduce the perinatal transmission of group B *streptococcus* (GBS) has resulted in a substantial decline in the incidence of EOS in high-income countries, to approximately 0.3–1 per 1000 live births (LBs) [3,4,5,6,7,8,9,10]. However, concerns have been raised regarding the potential of a subsequent increased risk of EOS due to non-GBS pathogens. In fact, previous studies have revealed increased rates of EOS due to *Escherichia coli* (*E. coli*) among pre-term and very-low birth weight (VLBW) infants [2,11,12,13,14,15,16,17,18]. Such a change would be alarming, as Gram-negative infections in newborns may present with high disease severity and case fatality rates [11,19,20,21]. The increase in IAP exposure of term and pre-term neonates has also raised concerns regarding the potential emergence of antimicrobial resistance in pathogens (including Gram-negative bacteria) to first-line, highly effective beta-lactam antibiotics. Finally, perinatal antibiotics may alter the development of the gut microbiota [22], and the consequent intestinal dysbiosis may affect the susceptibility later in life to diseases mediated by the immune system. Indeed, previous studies have demonstrated that the use of antibiotics during infancy correlates with the occurrence of asthma, atopic dermatitis, multiple sclerosis, and intestinal bowel diseases [23].

Thus, a better understanding of the changing epidemiology of EOS is needed to optimize prevention and treatment strategies. Changes in the burden of GBS and *E. coli* EOS—the two main pathogens causing EOS in high-income countries [3,5,9]—should be monitored over time. However, the number of cases at individual facilities remains low, preventing an accurate assessment of trends. Prospective area-based Italian data on EOS due to GBS or other pathogens are still lacking, although a large amount of data have been obtained on GBS invasive infections from a Northern region of Italy (Emilia-Romagna), a region where active area-based surveillance of GBS has been promoted since 2003 [24]. Starting from January 2016, the pre-existing GBS surveillance was associated with that of *E. coli*. For this reason, an area-based, five-year prospective exploratory study, including a birth cohort of approximately 160,000 LBs, was conducted to describe GBS- and *E. coli*-specific EOS incidence trends from 2016 to 2020. In addition, the characteristics of GBS and *E. coli* EOS, including antibiotic susceptibilities and outcomes, were compared.

## 2. Materials and Methods 

### 2.1. Study Design

An antenatal screening-based strategy for the prevention of GBS EOS [25,26] has been implemented in the entire region since 2003. The surveillance network includes eight microbiological laboratories, ten level 1 centres (<1000 LBs/year; inborn criteria: ≥2000 g, ≥35 weeks), four level 2 centres (>1000 LBs/year; inborn criteria: ≥1500 g, ≥32 weeks), and eight neonatal intensive care units (no restrictions for in- and out-born neonates). Prospective area-based surveillance for GBS and *E. coli* was carried out among LBs of all gestational ages (GAs) in a Northern region of Italy (Emilia-Romagna, ~four million people and 35,000 LBs/year). Patients were included if they had a blood or cerebrospinal fluid culture yielding GBS or *E. coli* within the first 3 postnatal days. To identify all cases, the prospective surveillance involved their reporting through a dual source. Microbiological laboratories were asked to report prospectively (from 1 January 2016 to 31 December 2020) each positive isolate from live-born infants. Clinicians were asked to complete a standardized form for each case of culture-proved EOS identified by the laboratory. In addition, to retrieve missed cases, all laboratories in the network were contacted at the end of each year for a detailed list of all registered cases. Incidence rates were calculated by using the number of live births, preterm deliveries (23–36 weeks’ gestation) and very low birth weight neonates (BW, <1500 g, 1501–2500 g, and >2500 g) provided by the Regional Health Agency. 

The Ethical Committee of the coordinating centre (Azienda Ospedaliero-Universitaria Policlinico di Modena; Prot. 910/2020) approved the project. Case reporting and isolate collection were determined to be non-research public health surveillance. To maintain patient confidentiality, spreadsheets submitted to the principal investigator were fully anonymous and did not include any identifiable data of patients or caregivers. Therefore, according to the policy of our ethics committee review board, patient consent was not required. The Strengthening the Reporting of Observational Studies in Epidemiology (STROBE) reporting guidelines were followed for this study. 

### 2.2. Clinical and Microbiological Practices

According to the individual centre policy, a single (1 mL) or double blood culture was obtained before antibiotic treatment from each neonate with clinically suspected EOS; some centres also obtained blood culture in asymptomatic infants due to maternal risk factors (RFs) for EOS (delivery at <37 weeks’ gestation; GBS bacteriuria; previous infant with GBS infection; rupture of membranes (ROM) ≥ 18 h before delivery; and intrapartum fever > 38 °C). Cerebrospinal fluid (CSF) culture was recommended in any case of suspected EOS but was deferred in unstable infants. Isolates were identified by MALDI-TOF MS, using Maldi Biotyper (Bruker Daltonics, Bremen, Germany) or Vitek-MS platforms (BioMérieux, Marcy l’Etoile, France). The capsular type of GBS isolates was identified by serological and molecular methods [27]. Antimicrobial susceptibility testing was performed using Vitek2 (BioMérieux, Marcy l’Etoile, France), MicroScan Walkaway (Beckman Coulter, Brea, CA, USA), or Phoenix (Becton Dickinson, Franklin Lakes, NJ, USA). Results were interpreted according to EUCAST clinical breakpoints. 

Ampicillin plus an aminoglycoside (usually gentamicin) was the initial empirical antimicrobial therapy recommended for suspected EOS. For each case of culture-proven EOS, maternal and neonatal information (both from delivery and case medical records) were detailed by surveillance officers using a standardized form. Maternal information included data on pre-natal GBS screening, delivery, IAP exposure, and RFs for EOS. Collected newborn data included infecting organism, antimicrobial susceptibility, antibiotic treatment, and outcome (e.g., death, brain lesions, age at discharge from hospital). 

### 2.3. Definitions

Antenatal screening: maternal vagino-rectal screening for GBS, performed within 5 weeks prior to delivery [25,26].Intrapartum antibiotic prophylaxis: intrapartum antibiotics administered for multiple reasons (e.g., GBS prophylaxis, suspected chorioamnionitis, maternal fever).Adequate IAP: penicillin, ampicillin, or cefazolin administered at least 4 h prior to delivery.Prolonged ROM: rupture of membranes ≥ 18 h before delivery.Pre-term neonates: neonates born at <37 weeks gestation.Late pre-term and full-term: neonates born at 34–36 and ≥37 weeks, respectively.Very low birth weight (VLBW) neonates: BW < 1500 g.EOS: yielding of GBS or *E. coli* in a blood culture obtained within the first three postnatal days [6,8,28].Asymptomatic bacteraemia: positive blood culture obtained due to maternal RFs for EOS in an infant who remained asymptomatic.Pneumonia: positive blood culture associated with respiratory distress syndrome and a characteristic radiographic appearance [9].Brain lesions: lesions confirmed by brain ultrasound or magnetic resonance imaging (MRI) in a newborn with clinical indications.Severe disease: any of the following: apnoea at onset; need for fluid resuscitation, catecholamine support, mechanical ventilation, or exchange transfusion; pneumonia; meningitis; brain lesions at hospital discharge; death.EOS-related death: death occurring within the first 7 postnatal days.

### 2.4. Statistical Analysis

Continuous variables are expressed using mean and standard deviation (SD) or median and interquartile range (IQR); binary and categorical data are reported as frequencies and percentages. The Student’s *t*-test was used for unadjusted comparisons of continuous variables between groups; Pearson’s χ^2^ test and Fisher’s exact test (when one or more cells had expected frequency < 5) were used for unadjusted comparisons of categorical variables between groups. Odds ratios (OR) and relative 95% confidence intervals (95% CI) were computed based on estimated parameters and variances from logistic regression models. Multi-variable logistic regressions were performed to adjust the results for GA.

## 3. Results

### 3.1. Overall and Pathogen Specific Incidence Rates, According to BW and GA

During the study period, 159,898 LBs were delivered at the participating centres (full-term neonates, *n* = 149,522; pre-term neonates, *n* = 10,322; missing information on GA, *n* = 54; BW ≤ 1500 g, *n* = 1568; BW 1501–2500 g, *n* = 9169; BW > 2500 g, *n* = 149,161). A total of 64 EOS cases were identified, giving a total incidence of 0.40 cases per 1000 LBs (95% CI 0.31–0.51). Of these, 39 were *E. coli* cases (0.24 cases per 1000 LBs) and 25 were GBS cases (0.16 cases per 1000 LBs). Thirty-nine (60.9%) of the 64 EOS cases were born full-term (GBS, *n* = 22; *E. coli n* = 17), and 53 of 64 neonates (82.8%) were Caucasians, while the remaining 11 neonates were Africans (*n* = 5), North Africans (*n* = 4), or others (*n* = 2).

During the 5-year surveillance period, the GBS- and *E. coli*-specific incidence rates, and their cumulative incidence, did not change (*p* = 0.99, *p* = 0.17, and *p* = 0.40, respectively; Figure 1). Overall, the rates of EOS among VLBW neonates were 40-fold higher compared to those with BW > 2500 g (OR 39.44, 95% CI 21.01–70.98; *p* < 0.001). EOS rates among VLBW neonates varied according to the pathogen, and *E. coli* rates were 16-fold higher than that of GBS among neonates with BW < 1500 g (*p* < 0.001). *E. coli* rates were also higher in neonates with BW 1501–2500 g (*p* = 0.03). In contrast, rates of *E. coli* and GBS EOS were comparable among infants with a BW > 2500 g (*p* = 0.35; Figure 2). Similarly, there was evidence of a trend toward increasing incidence among the low GA groups (Figure 3), with the lowest incidence reported in the interval 37–41 weeks’.

The demographics and characteristics of the entire population (overall, *E. coli*, and GBS cases) are summarized in Table 1. Cases of *E. coli* EOS were more likely to have a lower median BW and GA, as well as to be delivered with a VLBW. Their mothers exhibited a higher occurrence of histological chorioamnionitis, whereas prolonged ROM was borderline significant. Mothers of GBS EOS cases were more likely to present with a positive antenatal GBS screening, while cases of *E. coli* were more likely to be IAP-exposed and to receive an adequate IAP. Pre-term infants were more likely to be exposed to adequate IAP (pre-term neonates 50.0% vs. full-term neonates 21.1%; *p* = 0.018) and to be affected by *E. coli*. However, after adjustment for GA, neonates exposed to adequate IAP remained nine-fold more likely to be infected by *E. coli,* compared to GBS (OR 9.32, 95% CI 1.76–49.50; *p* = 0.009).

Among the overall cases of EOS, culture-proven meningitis was rare (two cases), but less than half of neonates in the entire cohort underwent a lumbar puncture. Three cases of *E. coli* EOS developed brain lesions. Two of these three (at 25 and 31 weeks of gestation, respectively) developed post-haemorrhagic hydrocephalus and focal periventricular white matter injuries, respectively. One infant was full-term (40 weeks of gestation) and developed a focal frontoparietal ischemic lesion. Four of five case fatalities were due to *E. coli,* three of which were ampicillin-resistant. Four of five who died were born pre-term (at 23, 23, 24, and 25 weeks of gestation, respectively). Among infants with GA ≥ 34 weeks, the risk of death was significantly lower (1/45, 2% vs. 4/19, 21%; *p* = 0.02). When adjusted for GA, the risk of death was 96% lower in IAP-exposed compared to unexposed infants, both including the whole cohort (OR 0.036, 95% CI 0.00–0.70; *p* = 0.03) or when considering only cases of *E. coli* EOS (95% lower in IAP-exposed compared to unexposed infants; OR 0.05, 95% CI 0.00–0.88; *p* = 0.04).

### 3.2. Factors Associated with Severe Disease

RFs associated with severe disease are shown in Table 2. From univariable models, we selected variables with *p*-value < 0.20. In the multi-variable analysis, BW and prolonged ROM were excluded due to potential problems of multi-collinearity, as suggested by the variance inflation factors (VIFs) (Table 3, Figure 4). The lack of IAP administration remained associated (*p* = 0.03) with severe disease (area under ROC curve = 0.80, H-L = 0.99, sensitivity 78%, specificity 71%).

### 3.3. Antimicrobial Susceptibility

Table 4 provides information concerning the antimicrobial susceptibility of pathogens. GBS isolates were universally susceptible to penicillin/ampicillin, vancomycin, and cefotaxime, while approximately one-third of them were resistant to clindamycin and erythromycin. Only eleven GBS isolates out of 64 EOS cases were available for further typing analysis. This small set comprised three strains resistant to erythromycin and clindamycin; two of them were serotype III and one was serotype Ib.

Most *E. coli* isolates (70%) were resistant to ampicillin. No significant differences were identified in ampicillin resistance between infants who received IAP or not (17 of 23 tested, 74% vs. 6 of 10 tested, 60%; *p* = 0.44). All *E. coli* isolates were susceptible to amikacin, 10% were resistant to gentamicin, and all were resistant to ampicillin. All *E. coli* isolates tested were susceptible to meropenem, while a minority were resistant to cefotaxime.

### 3.4. Comparison with Previous Emilia-Romagna Surveillance Reports

The GBS EOS incidence decreased, from 0.28 (95% CI 0.21–0.36) per 1000 LBs in 2003–2008 [29] to 0.16 (95% CI 0.10–0.23) per 1000 LBs in 2016–2020 (*p* = 0.01). Meanwhile, the E. coli EOS incidence increased from 0.13 (95% CI 0.08–0.21) in 2009–2012 [9] to 0.24 (95% CI 0.18–0.34) per 1000 LBs (*p* = 0.03) in 2016–2020, without any differences in BW and GA (data not shown). The E. coli case fatality rate significantly decreased from 37% (2009–2012) to 10% (2016–2020; *p* = 0.03), and E. coli ampicillin resistance remained stable (2016–2020: 70% vs. 2009–2012: 68%; *p* = 0.99).

## 4. Discussion

The incidence, microbiology, and mortality associated with GBS and *E. coli* EOS using Italian area-based surveillance data from 2016 to 2020 were considered in this study, thus updating information given in 2009–2012 [9] for the same area, where universal antenatal screening for GBS prevention has been greatly implemented over the years [30,31]. Compared to 2009–2012, cases of GBS EOS have significantly decreased (from 0.28 to 0.16 per 1000 LBs), as well as fatality rates associated to *E. coli* EOS. Cases of GBS slightly outweighed *E. coli* among full-term infants. Declines in GBS EOS have also been previously reported from other countries with robust screening policies in place [3,6,8,13,32,33,34]. However, unlike the result of a large surveillance study carried out in the USA [3], we observed a rise in the burden of *E. coli* EOS. We wonder whether rates of *E. coli* EOS have actually increased, as the prospective active surveillance in 2009–2012 did not cover *E. coli*, and cases were retrieved only from laboratory databases, through a retrospective study design [9].

The proportion of EOS cases due to GBS and *E. coli* tends to vary across studies, associated with the prevention strategy in place. GBS rates exceeded those of *E. coli* by at least two times in three recent European multi-centre studies from countries implementing a risk-based strategy [35,36,37]. In contrast, GBS slightly outweighs *E. coli* in countries (e.g., USA and Switzerland) where a screening-based strategy for GBS prevention has been greatly enhanced over the years [3,6,8]. With respect to our previous study [9], we found that *E. coli* is overtaking GBS as the leading pathogen causing EOS, although the 95% CIs around the incidence rates for GBS and *E. coli* overlapped. A similar finding has also been reported in a recent multi-centre non-area-based study in the USA [5]. As a result of the continuous decline in GBS EOS, *E. coli* is now emerging as an important pathogen among term infants. Our data reveal the urgent need for novel preventive strategies for *E. coli* EOS, addressing both pre-term and term infants. Similarly to GBS, high-risk women should be identified as a target for prevention [8]. Other investigators have found that ROM, especially in pre-term mothers, is more common in *E. coli* EOS than GBS [3]; thus, when EOS is suspected in pre-term neonates, prolonged ROM should immediately raise suspicion of *E. coli*.

Neonates exposed to adequate IAP had a nine-fold lower risk of GBS EOS, compared to *E. coli* EOS. Exposure to adequate IAP remained associated with *E. coli* EOS after controlling for GA. The overrepresentation of adequate IAP exposure among infants with *E. coli,* compared to those with GBS EOS, confirmed that beta-lactam IAP regimens (most typically penicillin or ampicillin) are effective in preventing GBS EOS, but are far less effective in preventing *E. coli* EOS. Furthermore, considering limitations due to the small number of fatal cases, exposure to IAP of any duration reduced the risk of death by more than 95% in the whole population of EOS cases, as well as when considering only *E. coli* infections, independently of GA. Finally, IAP of any duration remained associated with a lower risk of severe disease in the multi-variable analysis, after adjusting for GA, ethnicity, positive GBS screening, histological chorioamnionitis, and Apgar score. Apparently, IAP exposure reduces the severity of both GBS and *E. coli* EOS. This finding is in conflict with a previous systematic review, which excluded the significant effects of IAP on mortality from GBS and non-GBS infections [34]. Given the rarity of EOS-related deaths, the effects of IAP are difficult to assess. At present, it seems unethical to conduct randomized controlled trials comparing cohorts of neonates exposed or unexposed to IAP administration, as IAP is now a standard of care. However, the effects of IAP may be revealed through large cohort studies carried out in settings where adherence to IAP is strict.

Despite widespread use of IAP, GBS isolates remained universally susceptible to ampicillin, while *E. coli* resistance to ampicillin remained stable when compared to that in the period 2009–2012 [9]. Our data were consistent with those recently reported in the USA, where two-thirds of *E. coli* isolates responsible for EOS were ampicillin-resistant, particularly among IAP-exposed neonates [3]. In contrast, ampicillin resistance was comparable between IAP-exposed and -unexposed neonates in our study. Resistance to gentamicin was rare (10%) but notable, being always associated with ampicillin resistance. Our data were also in agreement with European surveillance data [38], where antimicrobial resistance was shown to remain at high levels (especially for gram-negative bacteria) during 2016–2020. Nevertheless, all our *E. coli* isolates tested were susceptible to meropenem.

As all GBS isolates tested were susceptible to ampicillin and 90% of *E. coli* isolates tested were susceptible to gentamicin, the recommended first-line antibiotics for EOS (ampicillin plus gentamicin) [39,40] seem appropriate. Nonetheless, it is a major concern that approximately 10% of *E. coli* isolates causing EOS were resistant to both ampicillin and gentamicin. Such resistance to both first-line antibiotics was consistent with data from the USA [5,41] and should be taken into consideration when neonates become worse despite the suggested empirical treatment. Broad-spectrum empirical antibiotics should be considered in selected, critically-ill neonates until culture results are available (i.e., VLBW infants born after prolonged ROM and exposed to prolonged courses of antepartum antibiotics) [5,9,40]. As all *E. coli* isolates tested were susceptible, amikacin could replace gentamicin in certain cases. Continued surveillance of trends in the pathogens responsible for EOS and their antimicrobial susceptibilities is warranted, in order to ensure that ampicillin and gentamicin remain appropriate empirical therapies.

This study had several limitations. First, cases of EOS were relatively few, possibly reducing the significance of some results. Indeed, the present study was intended to be exploratory research. Even though our results succeeded in identifying key associations and suggest further investigations, the results displayed were not corrected for multiple testing, potentially exposing the data to alpha error inflation. Secondly, due to its retrospective design, we were unable to retrieve complete information for all cases of EOS, and some data may have been biased. Finally, EOS cases were defined only in terms of positive blood cultures; this definition may be useful in epidemiological studies for benchmarking among centres, but gives insufficient information on the severity of sepsis.

## 5. Conclusions

*E. coli* was found to now be the leading pathogen in pre-term neonatal EOS, and its incidence was also close to that of GBS in full-term neonates. IAP was associated with a reduced risk of death and severe disease in GBS and *E. coli* EOS. Strategies for the prevention and management of *E. coli* EOS, especially among pre-term infants, should be considered. As the landscape of EOS microbiology and antimicrobial susceptibilities continues to evolve, ongoing surveillance of cases of *E. coli* and GBS infections is crucial.

## Figures and Tables

**Figure 1 microorganisms-10-01878-f001:**
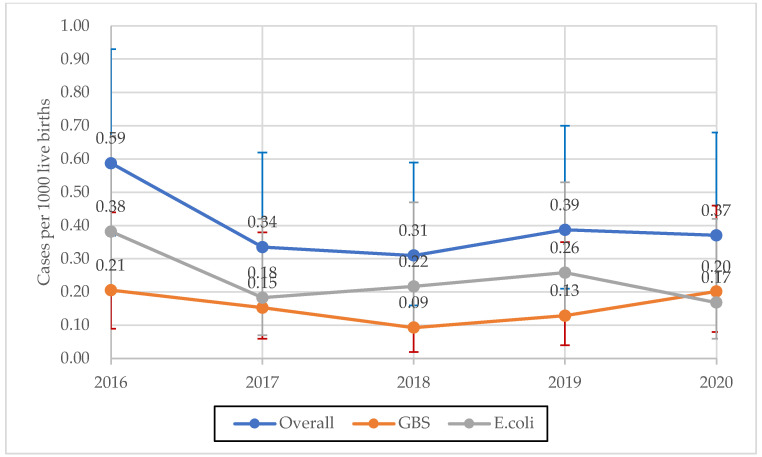
Annual incidence of overall, GBS, and *E. coli* early-onset sepsis; 2016 to 2020, Emilia-Romagna surveillance network.

**Figure 2 microorganisms-10-01878-f002:**
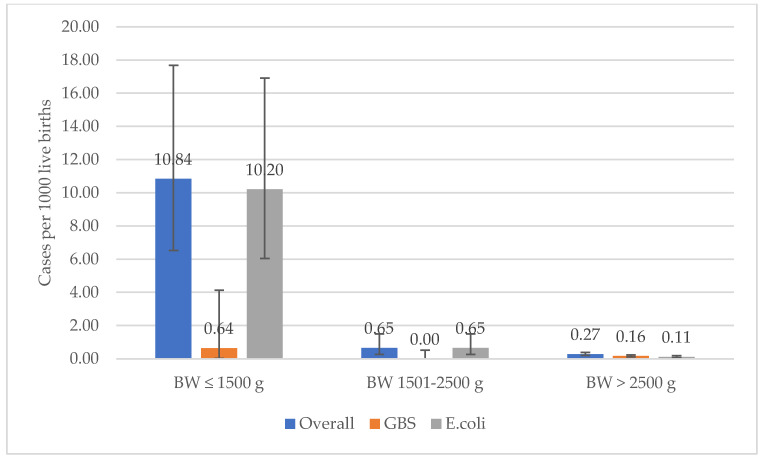
Incidence of GBS and *E. coli* early-onset sepsis by birth weight category; 2016 to 2020, Emilia-Romagna surveillance network.

**Figure 3 microorganisms-10-01878-f003:**
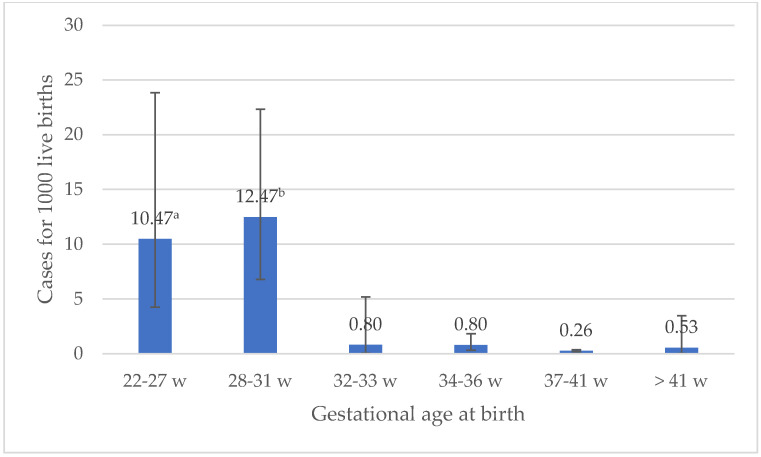
Early-Onset Sepsis incidence by gestational age at birth categories, 2016 to 2020 ^a^ Comparison of overall incidence between GA 22–27 w and higher GA groups (32–33 w, 34–36 w, 37–41 w, >41 w): *p* < 0.001 ^b^ Comparison of overall incidence between GA 28–31 w and higher GA groups (32–33 w, 34–36 w, 37–41 w, >41 w): *p* < 0.001.

**Figure 4 microorganisms-10-01878-f004:**
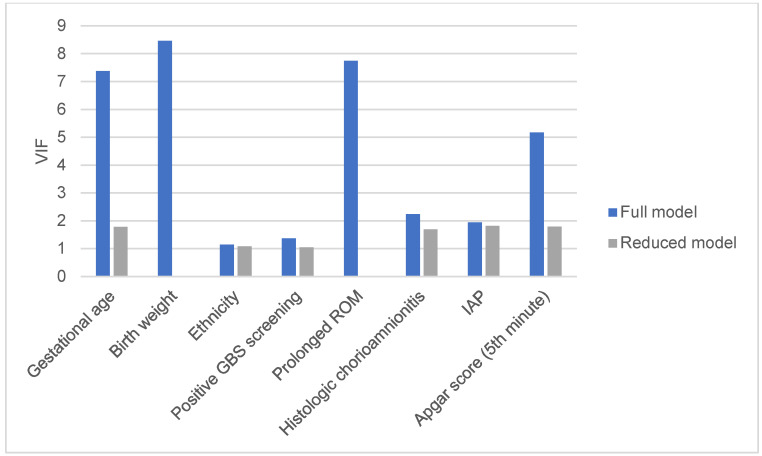
Comparison between variance inflation factors (VIFs) in the full model and those in the reduced model (excluding BW and prolonged ROM) in multi-variable analysis for severe disease.

**Table 1 microorganisms-10-01878-t001:** Demographics and clinical characteristics of newborns with early-onset sepsis (overall and according to pathogens).

Demographics and Clinical Characteristics	Overall (GBS and *E. coli*; *n* = 64)	*E. coli* (*n* = 39)	Missing	GBS (*n* = 25)	Missing	*p* ^a^
Positive antenatal GBS screening ^b^, *n* (%)	11 (20%)	3 (9%)	-	8 (36%)	-	0.04
Vaginal delivery, *n* (%)	40 (63%)	20 (51%)	-	20 (80%)	-	0.03
Prolonged ROM, *n* (%)	25 (39%)	19 (49%)	5	6 (24%)	8	0.05
Median duration of ROM, hours, (IQR)	17 (2–60)	20 (1–158)	8	9 (4–19)	8	0.02
Maternal intrapartum temperature ≥ 38 °C, *n* (%)	17 (27%)	10 (26%)	1	7 (28%)	-	0.99
Maternal GBS bacteriuria, *n* (%)	6 (9%)	5 (14%)	4	1 (5%)	5	0.40
Prior infant with GBS disease, *n* (%)	0 (0%)	-	-	0% (0%)	-	-
Clinical chorioamnionitis ^c^, *n* (%)	7 (11%)	4 (13%)	7	3 (14%)	4	0.99
Placental analysis performed, *n* (%)	30 (47%)	21 (64%)	18	9 (38%)	16	0.05
Histological chorioamnionitis, *n* (%)	18 (67%)	15 (47%)	7	3 (14%)	3	0.02
IAP exposure ^d^, *n* (%)	36 (56%)	27 (69%)	-	9 (36%)	-	0.009
Adequate IAP, *n* (%)	20 (31%)	18 (47%)	1	2 (8%)	1	0.002
Median birth weight, g (IQR)	2930 (1365–3479)	2235 (1120–3100)	-	3400 (3035–3842)	-	0.001
Birth weight < 1500 g, *n* (%)	17 (27%)	16 (41%)	-	1 (4%)	-	<0.001
Median gestational age, weeks (IQR)	38 (30–38)	35 (29–39)	-	40 (39–40)	-	0.001
Preterm neonates (<37 weeks gestation), *n* (%)	25 (39%)	22 (56%)	-	3 (12%)	-	0.001
Asymptomatic bacteraemia, *n* (%)	13 (20%)	6 (15%)	-	7 (28%)	-	0.22
Culture-proven meningitis ^e^, *n* (%)	2 (7%)	1 (6%)	-	1 (7%)	-	0.99
Pneumonia, *n* (%)	9 (14%)	3 (8%)	-	6 (24%)	-	0.14
Median length of hospital stay, d (IQR)	12 (9–30)	35 (29–39)	5	10 (8–15)	1	0.001
Brain lesions at discharge from hospital, *n* (%)	3 (5%)	3 (8%)	-	0 (0%)	-	0.28
Case fatalities, *n* (%)	5 (8%)	4 (10%)	-	1 (4%)	-	0.64

GBS, group B streptococcus; IAP, intrapartum antibiotic prophylaxis; IQR, interquartile range; ROM, rupture of membranes. ^a^
*p*-value for a difference between infants overall with GBS versus *E. coli,* by Fisher’s exact test or the χ^2^ test for categorical variables, or *t*-test for continuous variables. ^b^ Among 64 mothers, only 54 (84%; of which *E. coli n* = 32, GBS *n* = 22) underwent pre-natal GBS screening; percentages of positive cases were calculated only for mothers with an antenatal GBS screening. ^c^ Clinical suspicion of chorioamnionitis was documented in the medical record. ^d^ Intrapartum antibiotics were received for multiple reasons (GBS prophylaxis, suspected chorioamnionitis, caesarean delivery prophylaxis, maternal fever). ^e^ Among 64 neonates, only 30 (47%; of which *E. coli n* = 14, GBS *n* = 16) underwent lumbar puncture; percentages of meningitis were calculated only for cases with LP performed.

**Table 2 microorganisms-10-01878-t002:** Uni- and multi-variable analyses of factors associated with severe disease among infants with early-onset sepsis.

Characteristic	Univariable Analysis OR (95% CI)	*p*	Multi-Variable Analysis ^a^	*p*
Pathogen (*E. coli*)	1.49 (0.54–4.08)	0.44		
Gestational age	0.92 (0.84–1.01)	0.06	0.95 (0.80–1.13)	0.56
Birth weight	1.00 (0.99–1.00)	0.12		
Ethnicity	0.17 (0.02–1.59)	0.12	0.60 (0.04–9.36)	0.72
Sex	0.88 (0.33–2.36)	0.80		
Twin pregnancy	1.00 (0.06–16.71)	1.00		
Maternal age	1.02 (0.94–1.11)	0.63		
Positive GBS screening	0.37 (0.09–1.45)	0.15	0.58 (0.09–3.76)	0.56
GBS bacteriuria	1.04 (0.19–5.68)	0.96		
Intrapartum fever ≥38 °C	0.64 (0.21–1.98)	0.44		
Prolonged ROM	0.29 (0.10–0.85)	0.02		
Caesarean Section	1.00 (0.36–2.75)	0.95		
Clinical chorioamnionitis	0.53 (0.11–2.64)	0.44		
Histologic chorioamnionitis	2.91 (0.86–9.86)	0.09	2.24 (0.30–17.04)	0.44
IAP	0.46 (0.17–1.26)	0.13	0.12 (0.02–0.76)	0.03
Adequate IAP	0.61 (0.21–1.79)	0.61		
Apgar score (5th minute)	0.76 (0.59–1.00)	0.05	0.65 (0.37–1.17)	0.15
Ampicillin resistance	1.56 (0.53–4.56)	0.42		
Gentamicin resistance	0.84 (0.11–6.67)	0.87		
Resistance to first-line antibiotics ^a^	0.97 (0.13–7.33)	0.97		

IAP, intrapartum antibiotic prophylaxis; prolonged ROM, rupture of membranes ≥ 18 before delivery; VLBW, very low birth weight. ^a^ Combined resistance to both ampicillin and gentamicin.

**Table 3 microorganisms-10-01878-t003:** Variance inflation factors (VIFs) in full vs. reduced model (birth weight and prolonged ROM excluded).

	VIF (Full Model)	VIF (Reduced Model)
Gestational age	7.37	1.78
Birth weight	8.46	//
Ethnicity	1.15	1.08
Positive GBS screening	1.37	1.05
Prolonged ROM	7.74	//
Histologic chorioamnionitis	2.24	1.69
IAP	1.94	1.81
Apgar score (5th minute)	5.17	1.79

GBS, group B streptococcus; IAP, intrapartum antibiotic prophylaxis; prolonged ROM, rupture of membranes ≥ 18 h before delivery.

**Table 4 microorganisms-10-01878-t004:** Antimicrobial susceptibility of GBS and *E. coli* strains responsible for early-onset neonatal sepsis.

Susceptibility/Number of Isolates Tested (%) ^a^								
Pathogens	Amikacin	Ampicillin	Cefotaxime	Clindamycin	Erythromycin	Gentamicin	Meropenem	Vancomycin
GBS	-	24/24 (100%)	6/6 (100%)	15/22 (68%)	11/15 (73%)	-	-	22/22 (100%)
*E. coli*	39/39 (100%)	10/33 (30%)	34/37 (92%)	-	-	35/39 (90%)	33/33 (100%)	-

**^a^** Intermediate susceptibility was considered as resistance.

## Data Availability

The data that support the findings of this study are openly available (attachment to this submission).

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
