# Peer review of "Escherichia coli Is Overtaking Group B Streptococcus in Early-Onset Neonatal Sepsis"

_microorganisms, 2022, doi:10.3390/microorganisms10101878_

Round 1

Reviewer 1 Report

The manuscript by Francesca Miselli et al. describes very interesting Italian data on the bacterial presentation of EONS.

I read this manuscript with great interest and recommend many revisions according to my following comments:

Global: prefer passive voice.

Global: Numbers less than 12 should be written in capital letters.

Global: italicize all bacterial names/"versus".

Global: All acronyms should be introduced before their first use.

Methods: How was the number of subjects to be included in the study determined (as the authors have a "detailed list of all registered cases", it seems unacceptable that no consent was obtained)?

Methods: How was consent obtained from included patients?

Methods: Line 102: Please give details of maternal RF?

Methods Line 104: Please provide details of the platform used for MALDI-TOF identification.

Methods Line 118: Typing error "6,8,25".

Methods: How was the risk of alpha risk inflation due to multiple testing considered?

Results: A flow chart of included/selected patients/cases should be produced to facilitate understanding of the manuscript.

Results: Figure could not be printed/read? Conversion problem?

Results: How and when was the mother screened for GBS, as it is already well known that the methods and timing of screening can have a profound impact on positivity during birth?

Results: Antibiotic susceptibility of E. coli and GBS needs to be detailed as it can be critical to the management of pregnancies. For E. coli, although it is discussed, the authors should give details of K1 status, or discuss AST on this subject (see for example Proquot M. et al. 2021).

Results: "eTable 1" appears to be a typo.

Results: Section 3.8 should be placed in the discussion rather than the results section of the manuscript.

Discussion: Non-statistically significant trends should not be discussed or highlighted in the manuscript, as they detract from understanding the main message of the manuscript.

Discussion: The authors have stated that it is unethical to conduct randomized controlled trials on IAP. I disagree with this statement because it could be ethical and very interesting to compare different modalities of IAP (Amoxicillin vs. CoAmoxiclav for example). Please edit.

Author Response

We warmly thank the referees for their valuable suggestions, and certainly, our manuscript has improved after changes.

Below we reply to each point of both referee

Referee 1:

The manuscript by Francesca Miselli et al. describes very interesting Italian data on the bacterial presentation of EONS.

I read this manuscript with great interest and recommend many revisions according to my following comments:

  • Global: prefer passive voice.

The English reviewer was asked to change and prefer the passive voice.

  • Global: Numbers less than 12 should be written in capital letters. Changed as suggested when applicable

  • Global: italicize all bacterial names/"versus".. Changed as suggested

  • Global: All acronyms should be introduced before their first use. Changed as suggested

  • Methods: How was the number of subjects to be included in the study determined (as the authors have a "detailed list of all registered cases", For greater clarity, this is a large surveillance study, therefore we did not define an a priori number of cases to be included. All culture-confirmed cases of EOS (GBS and E. coli) during the study period were included in the study

  • Methods: it seems unacceptable that no consent was obtained)? How was consent obtained from included patients? We stated in the section Methods: The project was approved by the Ethical Committee of the coordinating centre (Azienda Ospedaliero-Universitaria Policlinico di Modena; Prot. 910/2020). The case reporting and isolate collection were determined to be non-research public health surveillance. To maintain patient confidentiality, spreadsheets submitted to the principal investigator were fully anonymous and did not include any identifiable data of patients or caregivers. Therefore, according to the policy of our ethics committee review board patient consent was exempted.

  • Methods: Line 102: Please give details of maternal RF? Maternal RFs (delivery at <37 weeks, GBS bacteriuria, a previous infant with GBS infection, rupture of membranes >18 h before delivery and intrapartum fever >38°C) are listed in Methods section and Table 1

  • Methods Line 104: Please provide details of the platform used for MALDI-TOF identification. Isolates were identified by MALDI-TOF MS, using Maldi Biotyper (Bruker Daltonics, Bremen, Germany) or Vitek-MS platforms (BioMérieux, Marcy l’Etoile, France). Changed as suggested

  • Methods Line 118: Typing error "6,8,25".                                                                                      Changed as suggested

  • Methods: How was the risk of alpha risk inflation due to multiple testing considered? The alpha risk inflation was not considered as a main issue in this work. Indeed, the present study was intended to be an exploratory research. Even if the results displayed were not corrected for multiple testing (and this can be a limitation of the work), they succeed in identifying key associations and in suggesting an approach to the problem for further investigations. We stated in the text “For this reason, we conducted an area-based 5-year prospective exploratory study”, and we added this limitation at the end of the manuscript

  • Results: A flow chart of included/selected patients/cases should be produced to facilitate understanding of the manuscript. For greater clarity, this is a large surveillance study, therefore we did not define an a priori number of cases to be included. All culture-confirmed cases of EOS (GBS and E. coli) during the study period were included in the study

  • Results: Figure could not be printed/read? Conversion problem? Figures were converted in power point and added as a supplementary file because their quality is lower compared to the original ones.

  • Results: How and when was the mother screened for GBS, as it is already well known that the methods and timing of screening can have a profound impact on positivity during birth? Study According to US guidelines (added in methods)

  • Results: Antibiotic susceptibility of E. coli and GBS needs to be detailed as it can be critical to the management of pregnancies. For E. coli, although it is discussed, the authors should give details of K1 status, or discuss AST on this subject (see for example Proquot M. et al. 2021). Unfortunately such an information is available only for 11 GBS cases, typed in detail at the National Health Institute (added in the text). Antibiotic susceptibility testing was  performed using Vitek2 (BioMérieux, Marcy l’Etoile, France) or MicroScan Walkaway (Beckman Coulter, Brea, CA, United States) or Phoenix (Becton Dickinson, Franklin Lakes, NJ, USA).

  • Results: "eTable 1" appears to be a typo. Changed as suggested

  • Results: Section 3.8 should be placed in the discussion rather than the results section of the manuscript. Introduction, Methods, Results and Discussion have been entirely rewritten

  • Discussion: Non-statistically significant trends should not be discussed or highlighted in the manuscript, as they detract from understanding the main message of the manuscript. . Changed as suggested

  • Discussion: The authors have stated that it is unethical to conduct randomized controlled trials on IAP. I disagree with this statement because it could be ethical and very interesting to compare different modalities of IAP (Amoxicillin vs. CoAmoxiclav for example). Please edit. . Edited as suggested

Reviewer 2 Report

The quality of the manuscript writing is very bad. Here are some suggestions to improve it. 

- The manuscript requires extensive English editing. 

- The introduction is very short and needs a lot of improvement.

- Names of the pathogens need to be in italic.

- In the study design section, add, how many samples were collected and talk about the sampling area.

- Add one table in section 2.1 to distribute your samples according to the history data. 

- I do not see any information on isolation and identification and procedures of AST.

- The presented data in this manuscript is not enough. For Example, You need to provide genotypic analysis and correlate it to the phenotypic analysis. 

Author Response

We warmly thank the referees for their valuable suggestions, and certainly, our manuscript has improved after changes.

Below we reply to each point of both referee

Referee 2:

The quality of the manuscript writing is very bad. Here are some suggestions to improve it. 

  • The manuscript requires extensive English editing. 

Edited by the Microorganism service (see the attached certificate)

  • The introduction is very short and needs a lot of improvement. Introduction, Methods, Results and Discussion have been entirely rewritten according to reviewers suggestions

  • Names of the pathogens need to be in italic.                                                    Changed as suggested

  • In the study design section, add, how many samples were collected and talk about the sampling area. For greater clarity, this is an area-based study; we added in the Methods section the number of microbiological laboratories involved in the surveillance (the number of neonatal centres had been already provided). It is hard to know the number of blood cultures obtained during the 5-year study period in the entire region. However, all culture-confirmed cases of EOS (GBS and E. coli) have been included in the study

  • Add one table in section 2.1 to distribute your samples according to the history data.  Given the low number of cases, we presented the EOS cases due to GBS and E. coli as incidence of cases per year (Fig 1)

  • I do not see any information on isolation and identification and procedures of AST. We stated in the section Methods “Antibiotic susceptibility testing was  performed using Vitek2 (BioMérieux, Marcy l’Etoile, France) or MicroScan Walkaway (Beckman Coulter, Brea, CA, United States) or Phoenix (Becton Dickinson, Franklin Lakes, NJ, USA).”

  • The presented data in this manuscript is not enough. For Example, You need to provide genotypic analysis and correlate it to the phenotypic analysis.  Unfortunately such an information is available only for 11 GBS cases, typed in detail at the National Health Institute (added in the text). Antibiotic susceptibility testing was  performed using Vitek2 (BioMérieux, Marcy l’Etoile, France) or MicroScan Walkaway (Beckman Coulter, Brea, CA, United States) or Phoenix (Becton Dickinson, Franklin Lakes, NJ, USA).

Round 2

Reviewer 1 Report

The manuscript has been greatly improved by the authors following my previous comments. 

However, some of them remain:

  • Methods: How was the number of subjects to be included in the study determined (as the authors have a "detailed list of all registered cases", For greater clarity, this is a large surveillance study, therefore we did not define an a priori number of cases to be included. All culture-confirmed cases of EOS (GBS and E. coli) during the study period were included in the study 

--> How did the authors determine the inclusion period needed to meet their objectives?

  • Methods: How was the risk of alpha risk inflation due to multiple testing considered? The alpha risk inflation was not considered as a main issue in this work. Indeed, the present study was intended to be an exploratory research. Even if the results displayed were not corrected for multiple testing (and this can be a limitation of the work), they succeed in identifying key associations and in suggesting an approach to the problem for further investigations. We stated in the text “For this reason, we conducted an area-based 5-year prospective exploratory study”, and we added this limitation at the end of the manuscript. 

--> Even though this is an exploratory study, the problem remains that the authors could conclude false positive results. I stand by my previous comments and request a review. 

  • Results: A flow chart of included/selected patients/cases should be produced to facilitate understanding of the manuscript. For greater clarity, this is a large surveillance study, therefore we did not define an a priori number of cases to be included. All culture-confirmed cases of EOS (GBS and E. coli) during the study period were included in the study.

--> A flow chart would be interesting to understand the percentage of patients included after application of the selection criteria, moreover, even in the absence of an a priori number of patients and considering a large selection, some files could be missing for example.

Author Response

We thank the reviewer for his valuable comments. Here we respond point by point to queries.

  • Methods: How did the authors determine the inclusion period needed to meet their objectives? We thank the referee for giving us an opportunity to further detail the study design (see methods section) and to improve the manuscript. This section has been rewritten. As stated, we carried out an active surveillance of GBS infections since 2003. During these years the prevention of EOD has been greatly implemented. Since 2016, we also started an active surveillance of E. coli cases. Therefore, by studying a population of approximately 160,000 newborns, over a sufficiently large time interval (5 years), we wonder whether the trend of GBS and E. coli early-onset sepsis have changed since the implementation of GBS prevention. The prospective, area-based study design is common in the pediatric literature and many studies have been published; if cohorts include thousands of infants, with an area-based denominator, the study generally allows fairly accurate estimates of disease incidence. However, we did not determine an a priori inclusion period to reach significant results, as it was not possible to know whether and how incidence rates would have changed.
  • Methods: Even though this is an exploratory study, the problem remains that the authors could conclude false positive results. I stand by my previous comments and request a review.

We further discussed the analysis with our statistician team: by definition, exploratory studies do not rely on prefixed hypothesis, thus they do not require correction. Since this is an exploratory study, such a correction would not be appropriate. We further acknowledged this limitation at the end of the discussion. 

  • Results: A flow chart would be interesting to understand the percentage of patients included after application of the selection criteria, moreover, even in the absence of an a priori number of patients and considering a large selection, some files could be missing for example.

Let us clarify again: we did not exclude any newborn. All patients with a blood or CSF culture yielding GBS or E. coli were included. Area-based study on EOS are usually designed in such a manner. Therefore, we would not know how to do "a flow chart of included/selected patients/cases". All neonates with positive blood or CSF culture have been included in the study (no missing cases; missing of cases was unlikely, since the prospective surveillance involved reporting through a dual source, clinicians AND laboratories). This was added in Methods. 

Reviewer 2 Report

However, you addressed all my comments. you need to provide a clean version of the manuscript after these substantial changes. it's hard to go through this version

Author Response

We thank the reviewer for his work. We attach a clean version of the final  manuscript. For greater clarity, we highlighted in yellow the changes of the second revision of our manuscript. Let us know if this is clear and if any other information is required. 
